# What are the perspectives of adults aged 18–40 living with type 2 diabetes in urban settings towards barriers and opportunities for better health and well-being: a mixed-methods study

Sarah Croke [1], Anna-Maria Volkmann [2], Catherine Perry [1], Ross A Atkinson [3], Alessio Pruneddu,[4] Lydia Morris,[5] Peter Bower [6]

For numbered affiliations see end of article.

**Correspondence to**
Dr Sarah Croke;
sarah.croke@manchester.ac.uk

## ABSTRACT

**Objectives** Delivered as part of the global assessment of diabetes in urban settings, this study explores different aspects of living with type 2 diabetes, for adults aged 18–40. Primary questions were as follows: (1) can we identify subgroups of adults under 40 years old sharing specific perspectives towards health, well-being and living with type 2 diabetes and (2) do these perspectives reveal specific barriers to and opportunities for better type 2 diabetes prevention and management and improved well-being?

**Design** The study employed a mixed-method design with data collected through demographic questionnaires, Q-sort statement sorting exercises, focus groups discussions and individual interviews.

**Setting** Primary care across Greater Manchester, UK.

**Participants** Those aged between 18 and 40, with a confirmed type 2 diabetes diagnosis, and living in Greater Manchester were eligible to participate. A total of 46 people completed the Q-sort exercise and 43 were included in the final analysis. Of those, 29 (67%) identified as female and 32 (75%) as white. Most common time since diagnosis was between 5 and 10 years.

**Results** The Q-sort analysis categorised 35 of the 43 participants (81%) into five subgroups. Based on average statement sorts for each subgroup, perspectives were characterised as: (1) stressed and calamity coping (n=13), (2) financially disadvantaged and poorly supported (n=12), (3) well-intentioned but not succeeding (n=5), (4) withdrawn and worried (n=2) and (5) young and stigmatised (n=3). Holistic analysis of our qualitative data also identified some common issues across these subgroups.

**Conclusions** Adults under 40 with type 2 diabetes are not a homogeneous group, but fall into five identifiable subgroups. They also experience issues specific to this age group that make it particularly difficult for them to focus on their own health. More tailored support could help them to make the necessary lifestyle changes and manage their type 2 diabetes better.

## INTRODUCTION

Type 2 diabetes is one of the most common metabolic disorders worldwide[1] and leads to

## STRENGTHS AND LIMITATIONS OF THIS STUDY

⇒ A key feature of a Q-study is the ability to reveal shared perspectives among a group of participants, and for those to be understood holistically.

⇒ The Q-sort requires a minimum of 35 participants to ensure statistical validity and a maximum of 64 (ie, not exceeding the number of statements).

⇒ The challenge to recruit younger adults living with type 2 diabetes as participants for this study were addressed by using multiple routes, including social media.

⇒ Restrictions due to the COVID-19 pandemic forced us to conduct all data collection via online platforms; this limited some from participating while providing opportunity to others.

⇒ Zabala's 'qmethod' analysis package allowed us to extract factors meeting statistical significance (high loading score on a single factor with $p<0.05$), and thereby establish shared perspectives in our sample despite sometimes having smaller numbers within some subgroups.

complications that cause both psychological and physical distress and put a huge burden on healthcare systems.[2] One response to this, in recent years, has been the Cities Changing Diabetes programme, a programme advocating for urgent action against type 2 diabetes on a global scale. Launched in 2015, via a partnership between Novo Nordisk, University College London (UCL) and Steno Diabetes Center, and implemented by City Partners, it aims to map the challenge, share solutions and act on the growing impact of type 2 diabetes in some of the world's great cities.[3] In 2019, Greater Manchester, UK became a City Partner.

A report conducted as part of this programme using the 'Rule of Halves' approach[4 5] identified patterns of diagnosis

and treatment in Greater Manchester and indicated that type 2 diabetes in the area is underdiagnosed in adults under 40 years of age. This also showed that, where type 2 diabetes was diagnosed, those under 40 seem less likely to receive appropriate care and achieve Glycated Haemoglobin (HbA1c) treatment targets when compared with older age groups. Building on this, we conducted the Urban Diabetes Priority Assessment to explore shared priorities, attitudes and points of view among local people with type 2 diabetes. Shared perspectives are important for informing future interventions and policies, while also helping to strengthen the global Cities Changing Diabetes research platform for understanding the social-cultural drivers of diabetes.[6]

Previous research suggests that adults aged 18–40 with type 2 diabetes have higher physical morbidity and mortality than other type 2 diabetes subgroups and poorer outcomes than their older counterparts.[7] People in this group are also more likely to be obese with suboptimal glycaemic control and have a high risk of developing at least one severe diabetes complication by the time they are in their 40s, reducing life expectancy by 15 years.[7] A further concern is poor adherence to medication.[8 9] Being a younger age at diagnosis is also associated with higher risk of mortality and vascular disease,[10] while people may require more intensive psychological and self-care support compared with those who develop type 2 diabetes later.[7] It is suggested that targeting younger patients, earlier in their trajectory, can present a unique opportunity to improve longer-term outcomes.[11]

We aimed to explore the following questions for adults under 40 with type 2 diabetes, living in Greater Manchester:

► Can we identify subgroups of adults under 40 years old sharing specific perspectives towards health, well-being and living with type 2 diabetes?
► Do these perspectives reveal specific barriers to and opportunities for better type 2 diabetes prevention and management and improved well-being?

## METHODS
### Study design
This study is part of the Cities Changing Diabetes protocol,[12] which forms part of a wider global mixed-methods approach.[13] This study draws on the principles of Q-methodology, a technique increasingly used by researchers to explore health-related decision-making and behaviours via the systematic and scientific study of subjective viewpoints. The aim of a Q-study is to extract as many world-views as possible, obtaining perspectives and point of views from highly informed individuals rather than from a sample representative of the general population, and employs a mixture of data collection and analyses to obtain insights.[14] It follows the earlier 'Rule of Halves' analysis and delivers the 'Urban Diabetes Priority Assessment'[15] which consists of two components.

The first component required participants to complete an online demographic questionnaire and computer based Q-sort, where participants prioritise a set of 64 statements according to their personal preference. The statements in the Q-sort reflect eight social factors and cultural determinants of diabetes vulnerability and were generated from prior Cities Changing Diabetes Vulnerability Assessment studies conducted in five global cities: Copenhagen, Mexico City, Houston, Shanghai and Tianjin. Across those sites, 746 individual assessments were carried out by A-MV and UCL colleagues. Each assessment lasted on average 2 hours and centred around a specific interview guide which had been translated and locally adapted. The assessments also captured demographic data via a survey as well as ethnographic observation data.[13] Examples of statements include: 'Healthy foods are a luxury', 'For me, it's easy to learn about diabetes', 'I tend to prioritise the needs of others over my own' and 'I think that diabetes is a death sentence' (full list in online supplemental appendix 1). The local research teams as well as the global team (UCL) shared consensus that they adequately reflected the lived experience of the participants in the Cities Changing Diabetes Vulnerability Assessment. The statement set was first validated as part of the Cities Changing Diabetes Vancouver Q-Sort study, and further validated as part of the Cities Changing Diabetes Moscow Q-sort Study.

The second component employed focus group discussions and individual interviews to facilitate in-depth exploration of the outcome of the Q-sort with selected participants.[6]

### Participants and recruitment
Adults aged between 18 and 40, with a confirmed type 2 diabetes diagnosis, and living in Greater Manchester were eligible to participate. The decision to focus recruitment on the 18–40 age group reflected findings from the Greater Manchester 'Rule of Halves' analysis. Participants were recruited through a combination of approaches: invitation by their general practitioner (GP) or pharmacy; invitation via a consent-to-contact database (Help BEAT Diabetes); articles in Diabetes UK newsletters; enlisting community leaders within South Asian communities and direct advertising on social media platforms (Facebook, Instagram and Twitter). Interested participants contacted researchers whereupon they were screened using a stratified sampling approach to ensure a diverse range of experiences, in regard to age, gender, ethnicity and socioeconomic status.

### Data collection
After giving informed consent, eligible participants were granted access to our online data collection tool (Qsortware).[16] Initially, participants completed a demographic survey (socioeconomic status, housing, employment, health insurance, long-term conditions, diabetes complications, height and weight). Next they undertook a statement sorting exercise ('Q-sort') to assess their perspectives on living with type 2 diabetes. Finally, those who had consented to be contacted were invited for stage

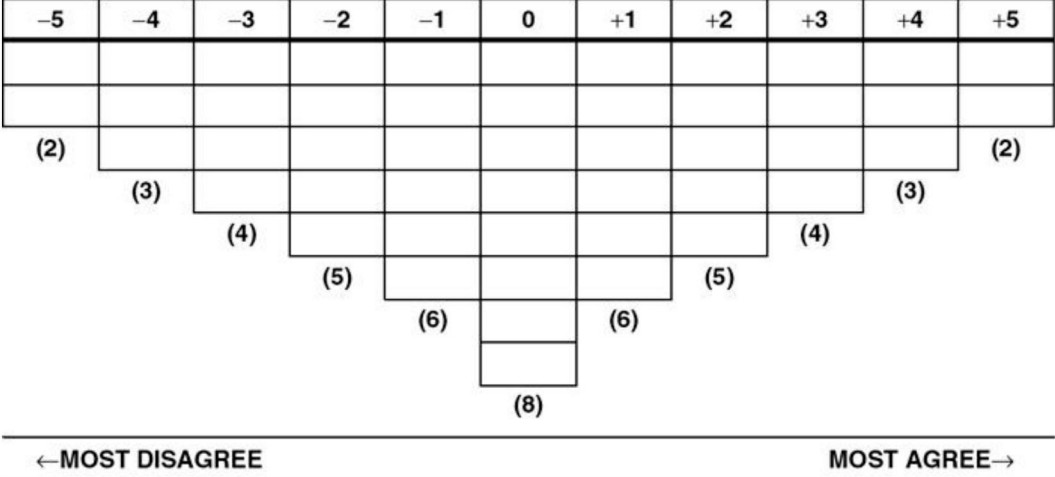

**Figure 1** Example of a forced distribution grid (Watts and Stenner,[20] p.17).

2 interviews and focus groups. Participants who agreed joined others to explore the results from the computer-based data collection, adding depth and detail to the statistical data, following a Cities Changing Diabetes-specific interview protocol.[14]

### Q-sort

The 64 statements had been synthesised from participant interviews in prior studies and represent a broad range of social and cultural factors relevant to health, well-being and type 2 diabetes.[13 14] The social factors were financial, time, resource, and geographical constraints and the cultural factors were traditions and conventions, health and illness, change and transition, self and other, and agency and choice.[3]

In the sorting exercise, participants were instructed to rank statements based on their individual level of agreement or disagreement with each statement. In the first step (preliminary sort), participants were asked to sort statements into three 'piles' (agree/disagree/neutral). In the second step (the final sort), participants added statements into a forced distribution grid (the Q-sort—see figure 1). We encouraged participants to add open-ended comments about each statement in a final step.

### Interviews and focus groups

The interviews and focus groups were led by an experienced qualitative health researcher (SC) with one or more other experienced researchers (A-MV, CP and RAA) in support, conducted remotely via secure video conferencing software (eg, Zoom or MS Teams), digitally recorded (audio only) and professionally transcribed verbatim. Topic guides covered the journey to diagnosis, type 2 diabetes management and the experiences of living with type 2 diabetes in Greater Manchester (see online supplemental appendix 2). We also explored areas where support was lacking and recommendations that could improve the experience of being a younger adult with type 2 diabetes. Focus Groups typically lasted 90 min and interviews lasted 60 min.

### Data analysis

We used Zabala's 'qmethod' analysis package,[17] run in R, to extract factors from the Q-sort and explore shared perspectives in our sample. To be assigned to a factor, participants needed a high loading score on a single factor, with $p<0.05$, and for this squared score to be higher than the squared sum of the scores obtained in all the other factors. Factors were described by the agreement, disagreement and neutrality expressed for each statement, both when compared with the other subgroup decision on that statement and in relation to the agreement/disagreement of the other statements of the Q-set, since there was a forced choice distribution involved. As is customary in Q-methodology, the research team (A-MV, AP, SC, CP and RAA) discussed various solutions and used the 'factor array' (see online supplemental appendix 1) to identify defining statements for each subgroup. A series of t-tests identified those statements that each subgroup sorted out differently, therefore, helping the interpretation. However, even those statements considered 'consensus' were still relevant, since they informed about issues/barriers universally perceived as significant by each perspective. 'Crib sheets' were used to extract the characteristics of each subgroup from (1) statements that ranked the highest in the factor compared with other factors; (2) statements that ranked the lowest in the factor compared with the other factors; (3) statements that ranked at −6 (very strongly disagree) and (4) statements that ranked at +6 (very strongly agree). Interpretation of the factors began by taking together all the statements in the 'strongly agree' area and those in the 'strongly disagree' area, especially when this configuration is in stark contrast with the other subgroups. The configuration of items and their inter-relationships provided the basis for initial interpretations of participant perspectives for each subgroup.

Thematic coding in NVivo V.12 and rapid assessment procedure sheets[18 19] were used to provide further, in-depth understanding of the issues that underpin

| Table 1 | Summary of participants (n=43) | |
|---|---|---|
| **Participant characteristic** | | |
| Age (years) | Mean | 35.3 |
| | SD | 4.4 |
| Sex | Male | 67% |
| | Female | 33% |
| Body mass index (BMI) (n=41) | Healthy weight (BMI 18.5–24.9) | 12.2% |
| | Overweight (BMI 25–29.9) | 7.3% |
| | Obese (BMI 30–39.9) | 41.5% |
| | Severely obese (BMI over 40) | 39.0% |
| Ethnicity | White | 74.4% |
| | Asian/Asian British | 20.9% |
| | Black/African/Caribbean/Black British | 2.3% |
| | Multiple ethnic groups | 2.3% |
| Household income (£) | <£30k | 46.5% |
| | ≥£30k | 53.5% |

the Q-sort and to supplement initial interpretations for each subgroup (see online supplemental appendix 3). These were synthesised with the demographic data and the open-ended comments from the Q-sort, to provide a comprehensive overview.

### Patient and public involvement

There was no patient involvement in the design or planning of the study. We made several attempts to recruit younger adults living with type 2 diabetes as patient and public contributors, to advise on delivery and dissemination of this study, however, this was unsuccessful within the timelines. This reflected the challenges of the COVID-19 pandemic and a general lack of engagement by this age group with formal health and community support services (eg, Diabetes UK; Help BEAT Diabetes; local community champions).

### RESULTS

From 46 completions, 43 participants provided analysable data in the Q-sort. Of these, 17 (40%) participated in a focus group or interview. Table 1 shows a summary of participant characteristics.

We extracted five factors (representing 81% of participants) to create our subgroups. Determining the best factor solution resulted from several discussions of the many variables involved, the complexity of the topic, and after excluding every factor solution with factors based on a single individual. We considered the total variance explained, the correlation between factors, the eigenvalues and reliability coefficients for each factor (Humphrey's rule). Individual Q-sorts were assigned to a factor according to specific criteria set by 'q-method', the library from R we used for data analysis. This activity,

called 'flagging', ensured each factor was highly distinguishable from the other ones extracted. All assigned participants achieved a high loading score on a single factor with p<0.05 and with the squared score higher than the squared sum of the scores obtained in all the other factors. We opted for a five factor solution since the variance explained was maximised in a stable statistical structure.

Based on their characteristics, we named these subgroups (1) stressed and calamity coping (n=13), (2) financially disadvantaged and poorly supported (n=12), (3) well intentioned but not succeeding (n=5), (4) withdrawn and worried (n=2) and (5) young and stigmatised (n=3). These interpretations reflect the key viewpoints captured by each study factor created from their relative and distinguishing statement rankings and demographic information, illustrated by the interview and focus group data.[20] They are woven and presented in a narrative style to present a seamless account of each subgroup's holistic viewpoint.

Holistic analysis of our qualitative data also identified some common issues across these five identifiable subgroups: overwhelming care obligations in those with young families; early or mid-career work pressures and stress; exclusion from social life; experiences with health professionals; and lack of specific skills and knowledge that could support them better. Some of these may be more relevant to adults aged 18–40 than other age groups and suggest a gap within current health and support services. Ethnicity had no clear influence on participant fit to subgroups. Household income showed potential influence in two of the subgroups: in subgroup three almost all participants reported incomes over £30k, while in subgroup four all participants reported incomes under £30k, a figure which is broadly in line with the current UK median household income.[21] Eight participants did not appear to fit any subgroup. These individuals still shared some common experiences with those in the five subgroups, including observations regarding ethnicity and household income, and two participated in qualitative interviews. Their inclusion in our holistic analysis provides additional interpretation and contextualisation.

### Subgroup 1: 'stressed and calamity coping' (n=13; qualitative interviews n=7)

These participants were mostly female (average age 34). Ten identified as white and three identified as Asian/Asian British. Their key viewpoint is that friends and family are very important. Despite initial challenges at diagnosis, they report having good awareness and some ability to manage their diabetes. They often ignore the onset of type 2 diabetes, and can be unclear about early symptoms.

> I don't know how long I was diabetic for, I wouldn't be surprised if I've been diabetic for at least a couple of years before my diagnosis. Now it's one of those cases

where you realise signs were there but they weren't as prominent (a male ages 20–30 years old, BMI 51)

They expressed a genuine desire to 'turn things around' but managing other demands (including stress, work and dependants) often made it difficult. Self-described 'food addiction' was also a feature.

Yeah, because I think food addiction is very real but there is no recognition for it really anywhere. (a female ages 30–40 years old, BMI 43)

### Subgroup 2: 'financially disadvantaged and poorly supported' (n=12; qualitative interviews n=4)

Participants in this subgroup were mostly female (average age 36). Ten identified as white, one identified as Asian/Asian British and one identified as Black/African/Caribbean/Black British. Their key viewpoint is they prioritise the needs of others, at the expense of themselves. They report lacking skills, finances and support to better manage their type 2 diabetes and may already have some complications. People are often 'prediabetic' for some time before being diagnosed with type 2 diabetes. Following diagnosis, they can feel unsure about how to manage their symptoms and do not always know how to find further information.

But there was nothing, it was just like right, well, no potatoes, come off the coke, do more exercise, see you later. (a female ages 30–40 years old, BMI 50)

They would like more support with diet; although time, skills and money for healthy eating and exercise are barriers. They also perceive a clinical focus on medication, despite struggling with side effects. They will often feel embarrassed, self-conscious and judged by others.

I haven't really told everybody either. I haven't told my partner; he doesn't know. I've told my mum but I think it's the same, it's that you feel like people are going to judge you for having it and it's your fault because of the way that you eat. (a female ages 30–40 years old, BMI 40)

### Subgroup 3: 'well-intentioned but not succeeding' (n=5; qualitative interviews n=3)

These participants were mostly male (average age 37). Three identified as white, one identified as Asian/Asian British and one identified as multiple ethnic groups. Their key viewpoint is they feel responsible for having diabetes, and so are responsible for managing it. They report being proactive in managing their health and trying to balance their diagnosis within their wider circumstances and lifestyles. Some regard being overweight as 'socially acceptable' and can have already developed associated complications. Often diagnosed as 'prediabetic' prior to full onset, they may report a family history of diabetes and expect a diagnosis sooner or later; although they are surprised to be diagnosed at such a young age.

It's my turn sooner or later it would be anyway. (a male ages 30–40 years old, BMI 29)

Family responsibilities are viewed as a positive motivation for change, and they feel that diet is more important than exercise in controlling type 2 diabetes. They like to ask a lot of questions and are quite optimistic about the future.

I think it's in moderation, and to be quite sensible with things, and what works for you. I've given up smoking, probably about five or six years too late, and again, because of my daughter, and priorities and things like that. (a male ages 30–40 years old, BMI 29).

### Subgroup 4: 'withdrawn and worried' (n=2; qualitative interviews n=0)

These participants were both male (average age 35). One identified as white and one identified as Asian/Asian British. Their key viewpoint is being isolated with concerns about future health. They find type 2 diabetes scary and say they would have tried to avoid it, if they had known more about it earlier. They are unsure whom to trust with their healthcare and/or advice. They find it hard to learn about their type 2 diabetes and do not feel well equipped to take care of themselves. They feel their choices are limited, increasing anxiety about their health. They find it hard to socialise (because of diabetes) which increases feelings of loneliness.

In the absence of interview and focus group data, this interpretation and identification of key viewpoint relies solely on their relative and distinguishing statement rankings, plus demographic information.

### Subgroup 5: 'young and stigmatised' (n=3; qualitative interviews n=1)

Participants were a mix of female and male (average age 33). Two identified as white and one identified as Asian/Asian British. Their key viewpoint is there is more to life than diabetes. They report being too young to worry too much about complications, although may have already experienced serious complications (eg, acute kidney injury). They feel embarrassment about having type 2 diabetes so young, and find it hard to accept their diagnosis.

One of the doctors were like, you can go blind and you can…and I'm like yeah, I'm not going to go blind, I'm not 60 yet. (a female ages 20–30 years old, BMI 24)

They believe type 2 diabetes can be reversed, with diet and exercise, but it is hard without the right environment and support. They struggle to tolerate side effects of medications, and choose non-adherence where they feel these are not compatible with their age and lifestyle. There is some resentment towards health services.

I was like I can't even socialise, I can't go even go out. And especially when I was younger I was like I'm not doing this. So, I stopped taking the injections at that point. (a female ages 20–30 years old, BMI 24)

### Common issues

In addition to identifying subgroups who share specific perspectives towards health, well-being and living with type 2 diabetes, we also identified common issues across the subgroups that create barriers to better type 2 diabetes prevention, management and improved well-being.

#### Overwhelming care obligations in those with young families

We learnt the busy nature of peoples' lives often meant that, whatever their different situations, making the best choices was not easy. People across all subgroups reported many barriers to lifestyle improvements. The combination of long working hours and childcare responsibilities made many people feel so tired at the end of the day that they would often reach for convenience foods. A lack of time, cooking skills or money were also reasons for less healthy choices. We saw signs of a gender split, with males largely citing family responsibilities as their motivation to 'do better', while females largely found these a barrier. Family life events and balancing responsibility for older and younger generations were a common source of stress.

#### Early or mid-career work pressures and stress

Many participants told us they struggled to accept their diagnosis at such a young age. They often had other priorities, such as socialising, progressing their careers and getting on the housing 'ladder'. Long working hours made it difficult to access exercise classes or gyms at quieter and cheaper times. For some, working patterns and environments made it difficult to take regular breaks, eat healthily or ask for more support. The workplace was viewed as somewhere people feel they could be better supported. A lack of employer support made it more difficult to adopt good type 2 diabetes management.

#### Exclusion from social life

A number of our participants spoke about feeling embarrassed or ashamed of being diagnosed with type 2 diabetes, or admitting their diagnosis to others. They often experienced unpleasant side effects from medication (flatulence, diarrhoea, etc) which could impact their social life, and being self-conscious about body size/shape could prevent access to exercise classes. Many expressed a desire to engage more with other people with type 2 diabetes, via support or exercise groups, suggesting they felt social support to be beneficial but lacking.

#### Experiences with health professionals

People shared with us a range of journeys to diagnosis, including family history, gestational diabetes and pancreatic injury. Some had a period of 'pre-diabetes' prior to diagnosis. Those having a type 2 diabetes diagnosis at a younger age felt that health professionals did not always listen to them or provide appropriate services or support, suited to their needs. Medication as a primary treatment was not popular with participants, yet few alternatives were presented when common medications caused intolerable side effects. This often led to refusal to take any prescribed medications. The perceived emphasis on services for older people with type 2 diabetes left many feeling abandoned or stigmatised. Previous negative experiences with health professionals could also influence current experiences, and contributed to feelings that younger people do not get as much support as older people. People in all subgroups felt they had not received enough education about 'being healthy' or recognising the 'signs of type 2 diabetes' prior to onset. The absence of formal support increased use of and dependency on internet sources, self-help books and anecdotal advice, which may not always be appropriate. A desire to 'reverse' or manage their type 2 diabetes with food and exercise, in the first instance, was met with a lack of support from health professionals. Health services were regarded as good quality once people were advanced into the 'complications' phase of type 2 diabetes, but poor in prevention and early management. Provision was perceived as better in surrounding towns and worse in the city.

### Lack of specific skills or knowledge that could support them better

We found that, despite their younger age, participants in our sample often had very high body mass index (BMI)—over 80% reporting data were classed as obese (BMI over 30) or severely obese (BMI over 40). However, many reported not having the skills or knowledge to manage their health alone. For those with young children this was a cause of additional concern, as it meant their children would be left to witness and/or deal with any crisis (hypo) episodes, and they were also concerned their children may repeat their habits. A number of people we spoke to had parents or older relatives who already had type 2 diabetes. The current provision of pre-diabetes and type 2 diabetes services was not felt to provide the skills or knowledge to meet the different needs of younger people. Younger adults across the subgroups noted how their finances and commitments and daily schedules are different to many older adults.

### DISCUSSION

This study aimed to identify and understand the perspectives and experiences of younger adults living with type 2 diabetes in Greater Manchester. We found five subgroups which illustrated the everyday challenges people face and the different strengths they may have to moderate these. These subgroups cut across socioeconomic factors such as household income, employment status, ethnicity, housing type and residence ownership. We also found common challenges related to this age group.

### Strengths and limitations

A key feature of a Q-study is the ability to reveal shared perspectives among a group of participants, and for those

to be understood holistically.[20] The Q-sort requires a minimum of 35 participants to ensure statistical validity and a maximum of 64 (ie, not exceeding the number of statements). We faced some challenges in recruiting younger adults living with type 2 diabetes as participants for this study, which thereby extended our recruitment period. This may reflect both the COVID-19 pandemic and a general lack of engagement by this age group with formal health services. To address this, we used multiple routes to optimise recruitment. In addition, the restrictions of the COVID-19 pandemic forced us to conduct all data collection via online platforms, which limited some from participating while providing opportunity to others. Zabala's 'qmethod' analysis package[17] allowed us to extract factors meeting statistical significance (high loading score on a single factor with $p < 0.05$), and thereby establish shared perspectives in our sample, despite sometimes having smaller numbers within some subgroups. Stratified sampling ensured we had a diverse range of experiences, with regard to age, gender, ethnicity and socioeconomic status, represented.

## Comparison with the wider literature

Previous literature focusing on type 2 diabetes with a younger adult age group have revealed some concerning data on trends and issues. To our knowledge, this is the first Q-sort study in this population.

Browne et al[7] have previously noted that younger adults are more likely to be obese with suboptimal glycaemic control and have a higher risk of developing at least one severe diabetes complication by their 40s. We found that people across all subgroups had very high BMI and that self-reported poor glycaemic control was common. Reasons for this varied between the subgroups, however, the multiple pressures, unacceptability of side effects and a lack of appropriate support were all felt to contribute.

Kunasegaran et al[8] and Nanayakkara et al[9] have both highlighted the issue and need to understand better the barriers patients face in regard to medication adherence, the impact of this on glycaemic control and how this can be improved. Browne et al[7] have also suggested that people aged ≤45 years may require more intensive psychological and self-care support compared with those who develop type 2 diabetes aged ≥45 years. From our study, poor toleration of medications and their side effects was a common issue raised by people across all subgroups. Others suggested a preference for non-pharmaceutical approaches (diet, counselling and exercise) as a primary approach but found little support from statutory services, and some simply felt they 'still had time' before needing to submit to control through medication.

Tan et al[11] and Nanayakkara et al[10] have previously called for effective targeting of younger patients, earlier in their journey, as longer type 2 diabetes duration increases risk of complications. We found that many people, including those with a family history of type 2 diabetes, did not know how to recognise early symptoms, or could easily confuse these with other causes. Those with a family history also believed it was to some extent inevitable they would develop type 2 diabetes, regardless of their habits, although only expected this when they were older. People without a family history were often ashamed and considered it their fault for developing type 2 diabetes, as they had not followed healthier lifestyles earlier. The lack of appropriate education, both in schools and in pre-diabetes courses, was felt to have contributed to this situation.

Walker et al[22] used a quantitative approach to examine the contribution of social determinants on glycaemic control in all adults (aged 18+) and found interventions that target self-efficacy and social support lead to improvements in glycaemic control, while clinical interventions can lower the burden of disease and address mutable social determinants. The overall age of people in our study was younger, however, we also found that greater social support and personalisation were proposed as potential solutions to common issues.

## Implications of the study

Across our subgroups, our findings suggest a perception of a lack of personalisation of diabetes support for those aged 18–40. People living with type 2 diabetes at a younger age want to be healthier, fitter, and more engaged in their own care. They frequently worried about their health problems and were anxious about the future. A lack of targeted programmes, that take their particular circumstances into account, make it harder to prioritise their type 2 diabetes care and achieve the necessary 'lifestyle modification' to prevent, delay or manage type 2 diabetes well.

The people we spoke to shared some common experiences/problems, but they each appeared to require different solutions. Younger people with type 2 diabetes need their GPs and care providers to recognise that they are not just individual patients, but are often parents and/or partners that live in complex and busy households. In addition, while much type 2 diabetes prevention and management is currently structured towards an older population, people who develop type 2 diabetes during their teens, twenties and thirties will ultimately need to manage this condition for longer, raising their risk of serious complications in the future. The condition is also heavily stigmatised and younger patients may suffer disproportionately from this.

As our primary purpose was to determine if we could identify subgroups of the 18–40 population who share perspectives, we did not specifically explore the impact of ethnicity or socioeconomic differences, with regard to the experience of the respondents or service access. We suggest further research could explore these issues.

## Conclusion

The Rule of Halves study that informed our work, as part of the Cities Changing Diabetes protocol, suggested that in Greater Manchester, younger adults are underdiagnosed, less likely to receive appropriate care and less

likely to achieve their HbA1c treatment targets compared with older age groups. Using Q-methodology, we have explored the experiences of younger adults with type 2 diabetes, in Greater Manchester, identifying discrete subgroups within this population and providing insights to different approaches that might help them avoid, delay onset or live well with type 2 diabetes. We also identify some common issues across the subgroups that may impact their long-term trajectory of type 2 diabetes and found particular patterns of lifestyle and social challenges that could inform services how to be better designed, to be sensitive to the needs and preferences of particular groups, including how these are advertised, delivered and assessed.

**Author affiliations**
[1]Division of Population Health, Health Services Research and Primary Care, The University of Manchester, Manchester, UK
[2]Department of Anthropology, University College London, London, UK
[3]Division of Nursing, Midwifery and Social Work, School of Health Sciences, Faculty of Biology, Medicine and Health, The University of Manchester, Manchester, UK
[4]Independent Research Consultant, Salford, UK
[5]Division of Psychology and Mental Health, School of Health Sciences, The University of Manchester, Manchester, UK
[6]NPCRDC, The University of Manchester, Manchester, UK

**Acknowledgements** The authors would like to thank Diabetes UK and Help BEAT Diabetes, hosted by Research for the Future (RftF): see https://www.ncbi.nlm.nih.gov/pmc/articles/PMC6805518/.

**Contributors** PB secured the funding. SC, A-MV, AP, CP, RAA and LM conducted the study and A-MV, AP, SC, CP and RAA designed and performed the analysis. SC drafted the paper and PB, A-MV, CP, RAA, AP and LM critically revised it. The corresponding author attests that all listed authors meet the relevant criteria and no other meeting these have been omitted. SC acts as the guarantor for this study.

**Funding** National Institute for Health and Care Research Applied Research Collaboration Greater Manchester (NIHR ARC-GM). Grant number NIHR200174.

**Disclaimer** The views expressed in this publication are those of the author(s) and not necessarily those of the National Institute for Health and Care Research or the Department of Health and Social Care.

**Competing interests** None declared.

**Patient and public involvement** Patients and/or the public were not involved in the design, or conduct, or reporting, or dissemination plans of this research.

**Patient consent for publication** Not applicable.

**Ethics approval** This study involves human participants and was approved by North-West—Greater Manchester Central Research Ethics Committee (REC reference: 21/NW/0030); HRA and Health and Care Research Wales (HCRW) IRAS project ID: 281050. Participants gave informed consent to participate in the study before taking part.

**Provenance and peer review** Not commissioned; externally peer reviewed.

**Data availability statement** Data are available on reasonable request.

**Author note** This research originates from work undertaken in the Cities Changing Diabetes (CCD) programme for Greater Manchester in partnership with and supported by Health Innovation Manchester and Novo Nordisk. CCD was launched in 2014 by the Steno Diabetes Center Copenhagen, University College London, and Novo Nordisk.

**ORCID iDs**
Sarah Croke http://orcid.org/0000-0001-6743-4573
Anna-Maria Volkmann http://orcid.org/0000-0002-5598-2967
Catherine Perry http://orcid.org/0000-0002-8496-6923
Ross A Atkinson http://orcid.org/0000-0001-8976-2754
Peter Bower http://orcid.org/0000-0001-9558-3349

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
