## [Reviewer comments · BMJ Open]

ARTICLE DETAILS

TITLE (PROVISIONAL)	What are the perspectives of adults aged 18-40 living with type 2 diabetes in urban settings towards barriers and opportunities for better health and wellbeing: a mixed-methods study
AUTHORS	Croke, Sarah; Volkmann, Anna-Maria; Perry, Catherine; Atkinson, Ross; Pruneddu, Alessio; Morris, Lydia; Bower, Peter

VERSION 1 – REVIEW

REVIEWER	Shaw, Ian University of Nottingham School of Sociology and Social Policy, Sociology and Social Policy
REVIEW RETURNED	07-Nov-2022

GENERAL COMMENTS	The paper raises some interesting questions and recommendations for service improvement. However, there was no reporting of ethnicity in the paper, which could have impacted both on the experience of the respondents and the findings with regards service access. Also I did not think the social class aspects (financially struggling) were sufficiently explored. The paper therefore raises some interesting points (albeit on a limited data set) but also leaves unanswered questions.
--

REVIEWER	Koetsenruijter, Jan Radboud Universiteit Nijmegen
REVIEW RETURNED	02-Dec-2022

GENERAL COMMENTS	Overall, I think this study addresses an interesting topic of younger people with type 2 diabetes. Also the used methods could provide interesting views on this population and how they perceive their health and life. However, I do have some concerns related to several sections in this manuscript. Introduction The reason why this paper focuses on age under 40 is well supported, but why sharing perspectives are of importance is not discussed at all. The introduction should provide a section on what sharing perspectives are and why they are important. This will also strengthen the choice to perform a Q-sorting methodology. Methods As the Q-sort method is a central element in this study, I would like to know more about how those statements were generated and/or selected. Is it an existing instrument or was it self-developed? By whom? Was it validated? How was made sure that all the social and cultural factors were represented in those statements? Also, the Q-sort statements stay a bit abstract. It would be good to
--

present some examples in the text so the reader knows what kind of statements were sorted. Also can be referred to appendix 1 in which all the statements are listed.

Results

Relating to the findings overall: make throughout the whole results section clear what findings are based on the Q-sorting and what comes from the interviews and focus groups. This is necessary for the reader to interpret and judge the provided evidence.

Relating to the Q-sorting cluster analysis: The theoretical process has been described in the methods which I think is sufficient. However, I would like to read here a bit more on how the practical process of defining these sub-groups went. Were other factor solutions considered? What decisions were discussed in the team? Why was this solution most satisfactory? Is there some kind of model fit, i.e. does this solution fit well to the data? This is important to me as those sub-groups are presented as facts and are the foundation of this paper.

Results sub-group 4: If no interviews were performed, where does this interpretation come from? Does sub-group 4 refer to Factor 4 in Appendix 1? I could not relate those statements to the text here. I would like to see clearly how the Q-sorting feeds into those results and interpretations.

Conclusion

"In Greater Manchester, younger adults are under-diagnosed, less likely to receive appropriate care and less likely to achieve their HbA1c treatment targets compared with older age groups."

This conclusion is not well supported by the findings. This population could be under-diagnosed, but this was not tested with an appropriate study design. Also, whether care is appropriate in comparison to older age groups was not tested. The conclusion should reflect their own findings and reflect the chosen methods.

Minor issues

The study design should explicitly be mentioned, not only the chosen methods.

The inclusion criteria "were eligible if they were over 18 years with diagnosed type 2 diabetes and lived in Greater Manchester" are mentioned twice.

" Finally, a sample took part in interviews and focus groups." How was this sample selected? What criteria were used?

Table 1: Add a short explanation how to interpret this cut off point for Household income in the UK setting. Median income? Poverty?

"These are potentially more relevant to adults aged 18-40 and suggest a lack of personalisation within current health and support services"  Very speculative. Rather for discussion.

In the strength and limitations the authors mention to have met statistical significance in their sample. I was wondering how this is to be interpreted as there was no random sample, but a purposefully. As far as I can see, statistical testing does not seem appropriate.

VERSION 1 – AUTHOR RESPONSE

Reviewer #1	
General remarks	
The paper raises some interesting questions and recommendations for service improvement. However, there was no reporting of ethnicity in the paper, which could have impacted both on the experience of the respondents and the findings with regards service access. Also I did not think the social class aspects (financially struggling) were sufficiently explored. The paper therefore raises some interesting points (albeit on a limited data set) but also leaves unanswered questions.	Thank you for highlighting this omission. We have added further observations about these demographic aspects on p.7 'Results' and reported ethnicity data in descriptions of the sub-groups. As our primary purpose was to determine if we could identify sub-groups of this population who share perspectives, we did not specifically explore the impact of ethnicity or socioeconomic differences, with regards to the experience of the respondents or service access. We agree these issues would benefit from further research, and have added text about this on p.12 'Discussion – implications of the study'.

Reviewer #2	
General remarks	
Overall, I think this study addresses an interesting topic of younger people with type 2 diabetes. Also the used methods could provide interesting views on this population and how they perceive their health and life. However, I do have some concerns related to several sections in this manuscript.	Thank you for this comment. We agree this is an important knowledge gap.
Additional points:	
1. Introduction: The reason why this paper focuses on age under 40 is well supported, but why sharing perspectives are of importance is not discussed at all. The introduction should provide a section on what sharing perspectives are and why they are important. This will also strengthen the choice to perform a Q-sorting methodology.	Thank you for highlighting this. We have added a justification for using Q-methodology and explained the importance of exploring shared perspectives for local and global progress in understanding Diabetes on p.4 'Introduction'. Please let us know if this is sufficient.
2. Methods: As the Q-sort method is a central element in this study, I would like to know more about how those statements were generated and/or selected. Is it an existing instrument or was it self-developed? By whom? Was it	Thank you for asking us to include further detail about this. We have added the information requested on p.5 'Methods: study design'.

validated? How was made sure that all the social and cultural factors were represented in those statements? Also, the Q-sort statements stay a bit abstract. It would be good to present some examples in the text so the reader knows what to kind of statements were sorted. Also can be referred to appendix 1 in which all the statements are listed.	Thank you for this suggestion. We have included 4 example statements in the text to help the reader to know what kind of statements were sorted, and have also referred readers to the full list in appendix 1.
3. Results: Relating to the findings overall: make throughout the whole results section clear what findings are based on the Q-sorting and what comes from the interviews and focus groups. This is necessary for the reader to interpret and judge the provided evidence.	Thank you for raising this issue regarding clarity. The sub-groups are the foundation of the paper, however an important aspect of Q-study is that the interpretation and key viewpoint of each sub-group is created from a synthesis of the relative and distinguishing statement rankings, demographic information, plus interview and focus group data. This is presented as a seamless narrative, to inform diabetes prevention and

Relating to the Q-sorting cluster analysis: The theoretical process has been described in the methods which I think is sufficient. However, I would like to read here a bit more on how the practical process of defining these sub-groups went. Were other factor solutions considered? What decisions were discussed in the team? Why was this solution most satisfactory? Is there some kind of model fit, i.e. does this solution fit well to the data? This is important to me as those sub-groups are presented as facts and are the foundation of this paper. Results sub-group 4: If no interviews were performed, where does this interpretation come from? Does sub-group 4 refer to Factor 4 in Appendix 1? I could not related those statements to the text here. I would like to see clearly how the Q-sorting feeds into those results and interpretations.	management interventions and contribute new knowledge about how sociocultural factors create specific barriers to, and opportunities for, successful diabetes prevention and better diabetes care and management. We have included text and a reference to explain this on p.7 'Results' and have clearly identified the key viewpoint of each sub-group throughout these narratives. We have also added further detail about the Q-sorting cluster analysis, on p.6 'Data analysis' and on p.8 'Results', in order to explain more about both the theory and the practical process of running the analysis, defining the sub-groups, considered solutions, team discussions and why the solution decided was the most satisfactory. We hope this addresses your concerns. Please let us know if this is sufficient. Thank you for suggesting we clarify this. We have added text to explain how we arrived at our interpretation, in the absence of interview data. Please let us know if this, combined with the explanation on p.7 about how interpretations are achieved, is satisfactory.
4. Conclusion: "In Greater Manchester, younger adults are under-diagnosed, less likely to receive appropriate care and less likely to achieve their HbA1c treatment targets compared with older age groups."	We apologies for the confusion. This statement refers to the Rule of Halves study that formed part of the Cities Changing Diabetes protocol and supported our focus on this specific population. We have amended the text

This conclusion is not well supported by the findings. This population could be under-diagnosed, but this was not tested with an appropriate study design. Also, whether care is appropriate in comparison to older age groups was not tested. The conclusion should reflect their own findings and reflect the chosen methods.	to remove this confusion and ensure the conclusion reflects our own findings and methods.
5. Minor issues: The study design should explicitly be mentioned, not only the chosen methods.	Thank you for this suggestion. We have added text to this effect on p.5 'Methods'. Please let us know if this is sufficient.

The inclusion criteria "were eligible if they were over 18 years with diagnosed type 2 diabetes and lived in Greater Manchester" are mentioned twice. " Finally, a sample took part in interviews and focus groups." How was this sample selected? What criteria were used? Table 1: Add a short explanation how to interpret this cut off point for Household income in the UK setting. Median income? Poverty? "These are potentially more relevant to adults aged 18-40 and suggest a lack of personalisation within current health and support services"  Very speculative. Rather for discussion. In the strength and limitations the authors mention to have met statistical significance in their sample. I was wondering how this is to be interpreted as there was no random sample, but a purposefully. As far as I can see, statistical testing does not seem appropriate.	Thank you for highlighting this duplication. We have corrected this. Thank you for asking us to expand on this. We have added the relevant information on p.5 'Data collection'. That you for suggesting this. We have added a short explanation on p.7 'Results'. Reading this back, we agree with your view. We have amended the text to read "Some of these may be more relevant to adults aged 18-40 than other age groups and suggest a gap within current health and support services." We think this is more accurate. You are correct there was no random sample. Statistical significance used here referred to the criteria for participants to be assigned to a factor or sub-group (i.e. achieving a high loading score on a single factor with $p < 0.05$ and for the squared score to be higher than the squared sum of the scores obtained in all the other factors). We have amended the text to clarify this use and avoid confusion.
--	---